# 3D-Aware Visual Question Answering about Parts, Poses and Occlusions

**Xingrui Wang**[1]   **Wufei Ma**[1]*   **Zhuowan Li**[1]†   **Adam Kortylewski**[2,3]   **Alan Yuille**[1]

[1] Johns Hopkins University   [2] Max Planck Institute for Informatics   [3] University of Freiburg

{xwang378, wma27, zli110, ayuille1}@jhu.edu     akortyle@mpi-inf.mpg.de

## Abstract

Despite rapid progress in Visual question answering (*VQA*), existing datasets and models mainly focus on testing reasoning in 2D. However, it is important that VQA models also understand the 3D structure of visual scenes, for example to support tasks like navigation or manipulation. This includes an understanding of the 3D object pose, their parts and occlusions. In this work, we introduce the task of 3D-aware VQA, which focuses on challenging questions that require a compositional reasoning over the 3D structure of visual scenes. We address 3D-aware VQA from both the dataset and the model perspective. First, we introduce Super-CLEVR-3D, a compositional reasoning dataset that contains questions about object parts, their 3D poses, and occlusions. Second, we propose PO3D-VQA, a 3D-aware VQA model that marries two powerful ideas: probabilistic neural symbolic program execution for reasoning and deep neural networks with 3D generative representations of objects for robust visual recognition. Our experimental results show our model PO3D-VQA outperforms existing methods significantly, but we still observe a significant performance gap compared to 2D VQA benchmarks, indicating that 3D-aware VQA remains an important open research area. The code is available at https://github.com/XingruiWang/3D-Aware-VQA.

## 1 Introduction

Visual question answering (*VQA*) is a challenging task that requires an in-depth understanding of vision and language, as well as multi-modal reasoning. Various benchmarks and models have been proposed to tackle this challenging task, but they mainly focus on 2D questions about objects, attributes, or 2D spatial relationships. However, it is important that VQA models understand the 3D structure of scenes, in order to support tasks like autonomous navigation and manipulation.

An inherent property of human vision is that we can naturally answer questions that require a comprehensive understanding of the 3D structure in images. For example, humans can answer the questions shown in Fig. 1, which ask about the object parts, their 3D poses, and occlusions. However, current VQA models, which often rely on 2D bounding boxes to encode a visual scene [2, 59, 25] struggle to answer such questions reliably (as can be seen from our experiments). We hypothesize this is caused by the lack of understanding of the 3D structure images.

In this work, we introduce the task of 3D-aware VQA, where answering the questions requires compositional reasoning over the 3D structure of the visual scenes. More specifically, we focus on challenging questions that require multi-step reasoning about the object-part hierarchy, the 3D poses of the objects, and the occlusion relationships between objects or parts.

---

*Wufei contributed to develop the 3D algorithm for multi-objects pose estimation.

†Zhuowan contributed to dataset construction, manuscript writing and conceptualizing the framework.

37th Conference on Neural Information Processing Systems (NeurIPS 2023).

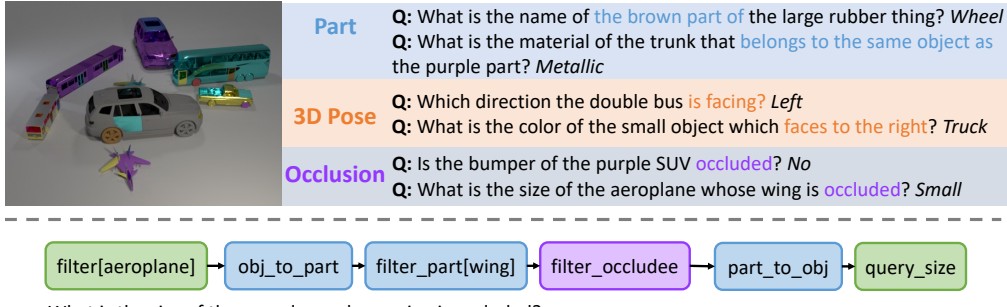

Figure 1: Examples from Super-CLEVR-3D. We introduce the task of 3D-aware VQA, which requires 3D understanding of the image, including the parts, 3D poses, and occlusions.

We address the challenging 3D-aware VQA task from both the dataset and the model perspective. From the dataset perspective, we introduce Super-CLEVR-3D, which extends the Super-CLEVR dataset [32] with 3D-aware questions. Given the visual scenes from Super-CLEVR that contain randomly placed vehicles of various categories, we define a set of 3D-aware reasoning operations and automatically generate 3D questions based on these operations. Fig. 1 shows examples of the images, questions and the underlying 3D operations for the questions. From the model perspective, we introduce PO3D-VQA, a VQA model that marries two powerful ideas: probabilistic neural symbolic program execution for reasoning and a deep neural network with 3D generative representations of objects for robust visual scene parsing. Our model first recovers a 3D scene representation from the image and a program from the question, and subsequently executes the program on the 3D scene representation to obtain an answer using a probabilistic reasoning process that takes into account the confidence of predictions from the neural network. We refer to our system as PO3D-VQA, which stands for **P**arts, Poses, and **O**cclusions in **3D V**isual **Q**uestion **A**nswering.

On Super-CLEVR-3D, we experiment with existing representative models, their variants, and our model PO3D-VQA. The results show that our model outperforms existing methods significantly, leading to an improvement in accuracy of more than 11%, which shows the advantage of the generative 3D scene parser and the probabilistic neural symbolic reasoning process. Moreover, further analysis on questions with different difficulty levels reveals that the improvements of our model are even greater on harder questions with heavy occlusions and small part sizes. Our results indicate that a reliable 3D understanding, together with the modular reasoning procedure, produces a desirable 3D-aware VQA model.

In summary, our contributions are as follows. (1) We introduce the challenging task of 3D-aware VQA and propose the Super-CLEVR-3D dataset, where 3D visual understanding about parts, 3D poses, and occlusions are required. (2) We propose a 3D-aware neural modular model PO3D-VQA that conducts probabilistic reasoning in a step-wise modular procedure based on robust 3D scene parsing. (3) With experiments, we show that 3D-aware knowledge and modular reasoning are crucial for 3D-aware VQA, and suggest future VQA methods take 3D understanding into account.

## 2   Related Work

**Visual Question Answering (VQA).** Rapid progress has been made in VQA [4] in both the datasets and the models. To solve the challenging VQA datasets [15, 61, 17, 45] with real images, multiple models are developed including two-stream feature fusion [2, 14, 28, 55, 23, 44, 30] or transformer-based pretraining [48, 36, 31, 59, 25]. However, the real datasets are shown to suffer from spurious correlations and biases [42, 16, 41, 1, 15, 26, 27]. Alternatively, synthetic datasets like CLEVR [24] and Super-CLEVR [32], are developed to study the compositional reasoning ability of VQA systems, which are also extended to study other vision-and-language tasks [34, 29, 53, 58, 6, 47, 20]. The synthetic datasets promote the development of neural modular methods [3, 54, 40, 22], where the reasoning is done in a modular step-by-step manner. It is shown that the modular methods have nice properties including interpretability, data efficiency [54, 40], better robustness [32] and strong performance on synthetic images [54]. However, most existing methods rely on region features [2, 59] extracted using 2D object detectors [46] for image encoding, which is not 3D-aware. We follow the works on the synthetic dataset and enhance the modular methods with 3D understanding.

**VQA in 3D.** Multiple existing works study VQA under the 3D setting, such as SimVQA [8], SQA3D [39], 3DMV-VQA [19], CLEVR-3D [51], ScanQA [52], 3DQA [52], and EmbodiedQA [13], which focus on question answering on the 3D visual scenes like real 3D scans [39, 51, 5, 52], simulated 3D environments [9, 13], or multi-view images [19]. PTR [20] is a synthetic VQA dataset that requires part-based reasoning about physics, analogy and geometry. Our setting differs from these works because we focus on 3D in the *questions* instead of 3D in the *visual scenes*, since our 3D-aware questions explicitly query the 3D information that can be inferred from the 2D input images.

**3D scene understanding.** One popular approach for scene understanding is to use the CLIP features pretrained on large-scale text-image pairs and segment the 2D scene into semantic regions [10, 43]. However, these methods lack a 3D understanding of the scene and cannot be used to answer 3D-related questions. Another approach is to adopt category-level 6D pose estimation methods that can locate objects in the image and estimate their 3D formulations. Previous approaches include classification-based methods that extend a Faster R-CNN model for 6D pose estimation [60, 38] and compositional models that predicts 6D poses with analysis-by-synthesis [38]. We also notice the huge progress of 3D vision language foundation models, which excel in multiple 3D vision-language understanding tasks [19, 37, 21]. Still, we focus on the reasoning with compositional reasoning that brings more interpretability and robustness [32].

## 3 Super-CLEVR-3D Dataset

To study 3D-aware VQA, we propose the Super-CLEVR-3D dataset, which contains questions explicitly asking about the 3D object configurations of the image. The images are rendered using scenes from the Super-CLEVR dataset [32], which is a VQA dataset containing synthetic scenes of randomly placed vehicles from 5 categories (car, plane, bicycle, motorbike, bus) with various of sub-types (*e.g.* different types of cars) and attributes (color, material, size). The questions are generated by instantiating the question templates based on the image scenes, using a pipeline similar to Super-CLEVR. In Super-CLEVR-3D, three types of 3D-aware questions are introduced: part questions, 3D pose questions, and occlusion questions. In the following, we will describe these three types of questions, and show the new operations we introduced for our 3D-aware questions about object parts, 3D poses, and occlusions. Examples of the dataset are shown in Fig. 1.

**Part questions.** While in the original Super-CLEVR dataset refers to objects using their holistic names or attributes, objects are complex and have hierarchical parts, as studied in recent works [33, 11, 20]. Therefore, we introduce part-based questions, which use parts to identify objects (*e.g.* "which vehicle has red door") or query about object parts (*e.g.* "what color is the door of the car"). To enable the generation of part-based questions, we introduce two new operations into the reasoning programs: `part_to_object`(·), which find the objects containing the given part, and `object_to_part`(·), which select all the parts of the given object. We also modify some existing operations (*i.e.* `filter`, `query` and `unique`), enabling them to operate on both object-level and part-level. With those reasoning operations, we collect 9 part-based templates and instantiate them with the image scene graph to generate questions.

**3D pose questions.** Super-CLEVR-3D asks questions about the 3D poses of objects (*e.g.* "which direction is the car facing in"), or the pair-wise pose relationships between objects (*e.g.* "which object has vertical direction with the red car"). The pose for an individual object (*e.g.* "facing left") can be processed in a similar way as attributes like colors, so we extend the existing attribute-related operations like `filter` and `query` to have them include pose as well. For pair-wise pose relationship between objects, we add three operations, *i.e.* `same_pose`, `opposite_pose` and `vertical_pose`, to deal with the three types of pose relationships between objects. For example, `opposite_pose`(·) returns the objects that are in the opposite pose direction with the given object. 17 templates are collected to generate 3D pose questions.

**Occlusion questions.** Occlusion questions ask about the occlusion between entities (*i.e.* objects or parts). Similar to 3D poses, occlusion can also be regarded as either an attributes for an entity (*e.g.* "which object is occluded"), or as a relationship between entities (*e.g.* "which object occludes the car door"). We extend the attribute-related operations, and introduce new operations to handle the pair-wise occlusion relationships: `filter_occludee` which filters the entities that are being occluded, `relate_occluding` which finds the entities that are occluded by the given entity, and `relate_occluded` which finds the entities that are occluding the given entity. Using these operations, 35 templates are collected to generate the occlusion questions.

# 4 Method

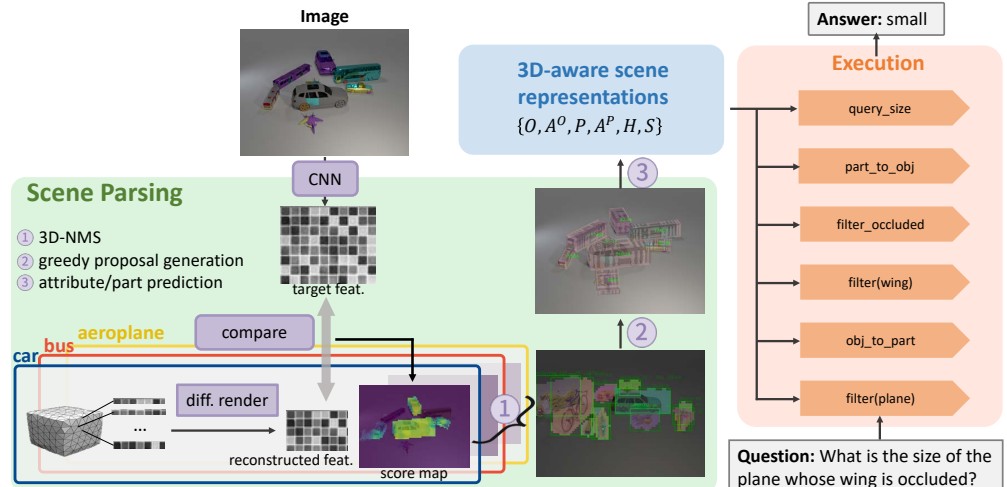

Figure 2: An overview of our model PO3D-VQA. The image is parsed into 3D-aware scene representations (blue box) using our proposed scene parser based on the idea of render-and-compare (green box). The question is parsed into a program composed of reasoning operations (orange box). Then the operations are executed on the 3D-aware scene representations to predict the answer.

In this section, we introduce PO3D-VQA, which is a parse-then-execute modular model for 3D-aware VQA. The overview of our system is shown in Fig. 2. We first parse the image into a scene graph representation that is aware of 3D information like object parts, 3D poses and occlusion relations, then we parse the question into a reasoning program and execute the program on the derived scene representations in a probabilistic manner. In Sec. 4.1, we define the scene representation required; in Sec. 4.2, we describe how we parse the image into the scene representation based on a multi-class 6D pose estimation model with non-trivial extensions; in Sec. 4.3, we describe how the question is executed on the derived scene representation to predict the answer.

## 4.1 3D-aware scene representation

Given an input image $I$, we parse it into a 3D-aware scene representation $R$ that contains the **objects** ($O$) with attributes ($A^o$), the **parts** ($P$) with attributes ($A^p$), the **hierarchical relationships** between objects and parts ($H$), and the **occlusion relationships** between them ($S$). The attributes include the 3D poses and locations of objects or parts, as well as their colors, materials, and sizes. The scene representation $R = \{O, P, A^o, A^p, H, S\}$ is comprehensive and therefore we can directly execute the symbolic reasoning module on this representation without taking into account the image any further.

In more detail, **objects** are represented as a matrix $O \in \mathbb{R}^{n \times N_{obj}}$ containing the probability scores of each object being a certain instance, where $n$ is the number of objects in the given image and $N_{obj}$ is the number of all possible object categories in the dataset (*i.e.* vocabulary size of the objects). Similarly, **parts** are represented as $P \in \mathbb{R}^{p \times N_{prt}}$, where $p$ is the number of parts in the image and $N_{prt}$ is the vocabulary size of the object parts. The **object-part hierarchy** is represented by a binary matrix $H \in \mathbb{R}^{n \times p}$, where $H_{ij} = 1$ if the object $i$ contains the part $j$ or $H_{ij} = 0$ otherwise. The attributes $A^o \in \mathbb{R}^{n \times N_{att}}$ and $A^p \in \mathbb{R}^{p \times N_{att}}$ containing probability scores of each object or part having a certain attribute or the value of bounding box. Here $N_{att}$ is the number of attributes including the 3D poses, location coordinates, colors, materials and sizes. **Occlusion relationships** are represented by $S \in \mathbb{R}^{(n+p) \times n}$, where each element $S_{ij}$ represents the score of object (or part) $i$ being occluded by object $j$.

## 4.2 Multi-class 6D Scene Parsing

While most existing VQA methods [2, 59] encode the image using pretrained object detectors like Faster-RCNN [46], we build our 6D-aware scene parser in a different way, based on the idea of analysis-by-synthesis through inverse rendering [49] which has the following advantages: first, the model prediction is more robust [49] as the render-and-compare process can naturally integrate a robust reconstruction loss to avoid distortion through occlusion; second, while the object parts

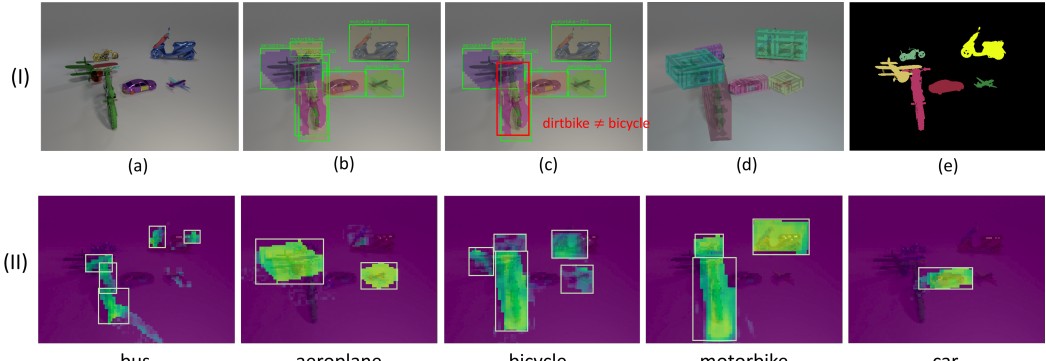

Figure 3: Visualization of intermediate steps in our scene parser. Given an image (a), per-category feature activation maps (shown in II) are computed through render-and-compare. Then the category-wise competition (3D-NMS) is performed (results shown in b) and a post-filtering step is taken to remove mis-detected objects (c). Based on the pose estimation results (d), we project the 3D object mesh back onto the image to locate parts and occlusions(e).

are usually very challenging for Faster-RCNN to detect due to their small size, they can be much easier located using the 3D object shape, by first finding the object and estimating its 3D pose, and subsequently locating the parts using the 3D object shape (as shown in our experimental evaluation).

However, we observe two open challenges for applying existing 6D pose estimators that follow a render-and-compare approach [38, 49]: (a) these pose estimators assume that the object class is known, but in Super-CLEVR-3D the scene parser must learn to estimate the object class jointly with the pose; and (b) the scenes in Super-CLEVR-3D are very dense, containing multiple close-by objects that occlude each other. In order to address these two challenges, we introduce several improvements over [38] that enable it to be integrated into a 3D-aware VQA model.

In the following, we first describe neural meshes [49, 38], which were proposed in prior work for pose estimation of *single objects* following an analysis-by-synthesis approach. Subsequently, we extend this method to complex scenes with densely located and possibly occluded objects to obtain a coherent scene representation, including object parts and attributes.

**Preliminaries.** Our work builds on and significantly extends Neural Meshes [38] that were introduced for 6D pose estimation through inverse rendering. The task is to jointly estimate the 6D pose (2D location, distance to the camera and 3D pose) of objects in an image. An object category is represented with a category-level mesh [49] $M_y = \{v_n \in \mathbb{R}^3\}_{n=1}^N$ and a neural texture $T_y \in \mathbb{R}^{N \times c}$ on the surface of the mesh $M_y$, where $c$ is the dimension of the feature and $y$ is the object category. Given the object 3D pose in camera view $\alpha$, we can render the neural mesh model $O_y = \{M_y, T_y\}$ into a feature map with soft rasterization [35]: $F_y(\alpha) = \Re(O_y, \alpha)$. Following prior work in pose estimation [49] we formulate the render-and-compare process as an optimization of the likelihood model:

$$p(F \mid O_y, \alpha_y, B) = \prod_{i \in \mathcal{FG}} p(f_i \mid O_y, \alpha_y) \prod_{i \in \mathcal{BG}} p(f_i' \mid B) \tag{1}$$

where $\mathcal{FG}$ and $\mathcal{BG}$ are the set of foreground and background locations on the 2D feature map and $f_i$ is the feature vector of $F$ at location $i$. Here the foreground and background likelihoods are modeled as Gaussian distributions.

To train the feature extractor $\Phi$, the neural texture $\{T_y\}$ and the background model $B$ jointly, we utilize the EM-type learning strategy as originally introduced for keypoint detection in CoKe[7]. Specifically, the feature extractor is trained using stochastic gradient descent while the parameters of the generative model $\{T_y\}$ and $B$ are trained using momentum update after every gradient step in the feature extractor, which was found to stabilize training convergence.

At inference time, the object poses $\alpha$ can be inferred by minimizing the negative log-likelihood w.r.t. the 3D pose $\alpha$ using gradient descent [38].

**Multi-object competition with 3D-NMS.** We extend Neural Meshes to predict the 6D object pose and class label in complex multi-object scenes. In particular, we introduce 3D-Non-Maximum-Suppression (3D-NMS) into the maximum likelihood inference process. This introduces a competition between Neural Meshes of different categories in explaining the feature map. In contrast to classical

2D-NMS, our 3D-NMS also takes into account the distance of each object to the camera and hence naturally enables reasoning about occlusions of objects in the scene.

We denote the 6D pose as $\gamma = \{x, l\}$, where $x = \{\alpha, \beta\}$ represents the 3D object pose $\alpha$ and object distance to the camera $\beta$, and $l$ is the 2D object location in the feature map. We first detect the 6D poses of each object category independently and apply 2D-NMS such that for each 2D location $l'$ in a neighborhood defined by radius $r$, the predicted 6D pose $\{x, l\}$ yields the largest activation:

$$\max_x p(F \mid x, l) \ \ s.t. \ \ p(F \mid x, l) > p(F \mid x, l'), \ \ \forall l' \in \{l' \mid 0 < |l' - l| < r\} \tag{2}$$

We enable multi-category 6D pose estimation by extending this formulation to a 3D non-maximum suppression (3D-NMS). Using $\mathcal{Y}$ to represent the set of all object categories, we model the category label $y$ from a generative perspective:

$$\max_x p(F \mid x, l, y) \ \ s.t. \ \ p(F \mid x, l, y) > p(F \mid x, l', y), \ \ \forall l' \in \{l' \mid 0 < |l' - l| < r\} \tag{3}$$

$$and \ \ p(F \mid x, l, y) > p(F \mid x, l, y'), \ \ \forall y' \neq y \in \mathcal{Y} \tag{4}$$

**Dense scene parsing with greedy proposal generation.** Typically, object detection in complex scenes requires well chosen thresholds and detection hyperparameters. Our render-and-compare approach enables us to avoid tedious hyperparameter tuning by adopting a greedy approach to maximize the model likelihood (Eq. (1)) using a greedy proposal strategy. In particular, we optimize the likelihood greedily by starting from the object proposal that explains away the most parts of the image with highest likelihood, and subsequently update the likelihood of the overlapping proposals taking into account, that at every pixel in the feature map only one object can be visible [56]. Formally, given a list of objects proposals $\{o_i = (O_{y,i}, \alpha_{y,i})\}_{i=1}^k$ (with predicted category label $y$ and 6D pose $\alpha$), we first order the object proposals based on their likelihood score $s = p(F \mid o_i, B)$ such that $s_i \leq s_j$ for $i < j$. Based on the ordering, we greedily update the 6D pose $\alpha_j$ and the corresponding proposal likelihood for object $o_j$ by masking out the foreground regions of previous objects $o_i$ with $1 \leq i \leq j - 1$. In this way, we can largely avoid missing close-by objects or duplicated detection.

**Part and attribute prediction.** Given the predicted location and pose of each object, we project the object mesh back onto the image to get the locations for each part. To predict the attributes for the objects and parts, we crop the region containing the object or part from the RGB image, and train an additional CNN classifier using the cropped patches to predict the attributes (color, size, material) and the fine-grained classes (*i.e.* different sub-types of cars) of each patch using a cross-entropy loss. The reason why this additional CNN classifier is needed instead of re-using the features from the 6D pose estimator is that the pose estimation features are learned to be invariant to scale and texture changes, which makes it unsuitable for attribute prediction.

**Post-filtering.** Finally, we post-process the located objects using the fine-grained CNN classifier. We compare the category labels predicted by the 6D pose estimator with the ones predicted by the CNN classifier, and remove the objects for which these two predictions do not agree. This post-filtering step helps with the duplicated detections that cannot be fully resolved with the 3D-NMS.

**Summary.** Fig. 2 provides an overview of our scene parser and Fig. 3 visualize the intermediate results. With the idea of render-and-compare (shown in the green box of Fig. 2), the model first computes an activation map for each possible object category (Fig. 3II). Next, to infer the category for each object, the category-wise competition 3D-NMS is performed (Fig. 3b) and a post-filtering step is taken to remove mis-detected objects (Fig. 3c). Fig. 3d shows the 6D pose estimation results. To predict parts, we project the 3D object mesh back onto the image to locate parts based on projected objects (Fig. 3e). In this way, the input image can be parsed into a 3D-aware representation, which is ready for the question reasoning with program execution.

### 4.3 Program execution

After the 3D-aware scene representations are predicted for the given image, the question is parsed into a reasoning program, which is then executed on the scene representation to predict the answer. The question parsing follows previous work [54], where a LSTM sequence-to-sequence model is trained to parse the question into its corresponding program. Like P-NSVQA [32], each operation in the program is executed on the scene representation in a probabilistic way. In the following, we describe the execution of the new operations we introduced.

The part-related operators are implemented by querying the object-part hierarchy matrix $H$, so that the object containing a given part (`part_to_object`) and the parts belonging to the given object

(`object_to_part`) can be determined. The pose-related operators are based on the estimated 3D pose in the object attributes $A^o$. For the `filter` and `query` operations regarding pose, the 3D poses are quantified into four direction (left, right, front, back). For the pair-wise pose relationships, the azimuth angle between two objects is used to determine the same/opposite/vertical directions. The occlusion-related operations are implemented by querying the occlusion matrix $S$. Based on the occlusion scores $S_{ij}$ representing whether entity $i$ being occluded by entity $j$, we can compute the score of one entity being occluded $\sum_j S_{ij}$ (`filter_occludee`), find the entities that occlude a given entity (`relate_occluded`), or find the entities that are occluded by a given entity (`relate_occluded`).

## 5 Experiments

### 5.1 Evaluated methods

We compare our model with three representative VQA models: FiLM [44], mDETR [25], and PNSVQA [32]. Additionally, we introduce a variant of PNSVQA, PNSVQA+Projection, to analyze the benefit of our generative 6D pose estimation approach.

**FiLM [44]** *Feature-wise Linear Modulation* is a representative two-stream feature fusion method. The FiLM model merges the question features extracted with GRU [12] and image features extracted with CNN and predicts answers based on the merged features.

**mDETR [25]** mDETR is a pretrained text-guided object detector based on transformers. The model is pretrained with 1.3M image and text pairs and shows strong performance when finetuned on downstream tasks like referring expression understanding or VQA.

**PNSVQA [32]** PNSVQA is a SoTA neural symbolic VQA model. It parses the scene using MaskR-CNN [18] and an attribute extraction network, then executes the reasoning program on the parsed visual scenes with taking into account the uncertainty of the scene parser. To extend PNSVQA to the 3D questions in Super-CLEVR-3D, we add a regression head in the attribute extraction network to predict the 3D posefor each object; parts are detected in a similar way as objects by predicting 2D bounding boxes; the part-object associations and occlusions are computed using intersection-over-union: a part belongs to an intersected object if the part label matches the object label, otherwise it is occluded by this object.

**PNSVQA+Projection** Similar with NSVQA, this model predicts the 6D poses, categories and attributes using MaskRCNN and the attribute extraction network. The difference is that the parts and occlusions are predicted by projecting the 3D object models onto the image using the predicted 6D pose and category (same with how we find parts and occlusions in our model). This model helps us ablate the influence of the two components in our model, *i.e.* 6D pose prediction by render-and-compare, and part/occlusion detection with mesh projection.

### 5.2 Experiment setup

**Dataset.** Our Super-CLEVR-3D dataset shares the same visual scenes with Super-CLEVR dataset. We re-render the images with more annotations recorded (camera parameters, parts annotations, occlusion maps). The dataset splits follow the Super-CLEVR dataset, where we have 20k images for training, 5k for validation, and 5k for testing. For question generation, we create 9 templates for part questions, 17 templates for pose questions, 35 templates for occlusion questions (with and without parts). For each of the three types, 8 to 10 questions are generated for each image by randomly sampling the templates. We ensure that the questions are not ill-posed and cannot be answered by taking shortcuts, *i.e.* the questions contain no redundant reasoning steps, following the no-redundancy setting in [32]. More details including the list of question templates can be found in the Appendix.

**Implementation details.** We train the 6D pose estimator and CNN attribute classifier separately. We train the 6D pose estimator (including the contrastive feature backbone and the nerual mesh models for each of the 5 classes) for 15k iterations with batch size 15, which takes around 2 hours on NVIDIA RTX A5000 for each class. The attribute classifier, which is a ResNet50, is shared for objects and parts. It is trained for 100 epochs with batch size 64. During inference, it takes 22s for 6D pose estimation and 10s for object mesh projection for all the objects in one image. During inference of the 6D pose estimator, we assume the theta is 0. During 3D NMS filtering, we choose the radius $r$ as 2, and we also filter the object proposals with a threshold of 15 on the score map.

### 5.3 Quantitative Results

We trained our model and baselines on Super-CLEVR-3D's training split, reporting answer accuracies on the test split in Tab. 1. Accuracies for each question type are detailed separately.

Table 1: Model accuracies on the Super-CLEVR-3D testing split, reported for each question type, *i.e.* questions about parts, 3D poses, occlusions between objects, occlusions between objects and parts.

|  | Mean | Part | Pose | Occ. | Part+Occ. |
|---|---|---|---|---|---|
| FiLM [44] | 50.53 | 38.24 | 67.82 | 51.41 | 44.66 |
| mDETR [25] | 55.72 | 41.52 | 71.76 | 64.99 | 50.47 |
| PNSVQA [32] | 64.39 | 50.61 | **87.78** | 65.80 | 53.35 |
| PNSVQA+Projection | 68.15 | 56.30 | 86.70 | 70.70 | 58.90 |
| **PO3D-VQA (Ours)** | **75.64** | **71.85** | 86.40 | **76.90** | **67.40** |

**Comparison with baselines.** First, among all the baseline methods, the neural symbolic method PNSVQA performs the best (64.4% accuracy), outperforming the end-to-end methods mDETR and FiLM by a large margin ($> 8\%$). This shows the advantage of the step-wise modular reasoning procedure, which agrees with the findings in prior works that the modular methods excel on the simulated benchmarks that require long-trace reasoning. Second, our model achieves 75.6% average accuracy, which significantly outperforms all the evaluated models. Especially, comparing our PO3D-VQA with its 2D counterpart NSVQA, we see that the injection of 3D knowledge brings a large performance boost of 11%, suggesting the importance of the 3D understanding.

**Comparison with PNSVQA variants.** By analyzing the results of PNSVQA variants (*PNSVQA*, *PNSVQA+Projection*, and our *PO3D-VQA*), we show (a) the benefit of estimating object 3D poses using our analysis-by-synthesis method over regression and (b) the benefit of object-part structure knowledge. First, by detecting part using 3D model projection, *PNSVQA+Projection* improves the *PNSVQA* results by 4%, which indicates that locating parts based on objects using the object-part structure knowledge is beneficial. Second, by estimating object 6D poses with our generative render-and-compare method, our *PO3D-VQA* outperforms *PNSVQA+Projection* by 7% (from 68.2% to 75.6%), showing the advantage of our render-and-compare model. Moreover, looking at the per-type results, we find that the improvement of our PO3D-VQA is most significant on the part-related questions (21% improvement over PNSVQA) and part-with-occlusion questions (14%), while the accuracy on pose-related questions does not improve. The reason is that part and occlusion predictions require precise pose predictions for accurate mesh projection, while the pose questions only require a rough pose to determine the facing direction.

### 5.4    Analysis and discussions

To further analyze the advantage of PO3D-VQA over other PNSVQA variants, we compare the models on questions of different difficulty levels. It is shown that the benefit our model is the most significant on hard questions. In Fig. 4, we plot the relative accuracy drop [3] of each model on questions with different occlusion ratios and questions with different part sizes.

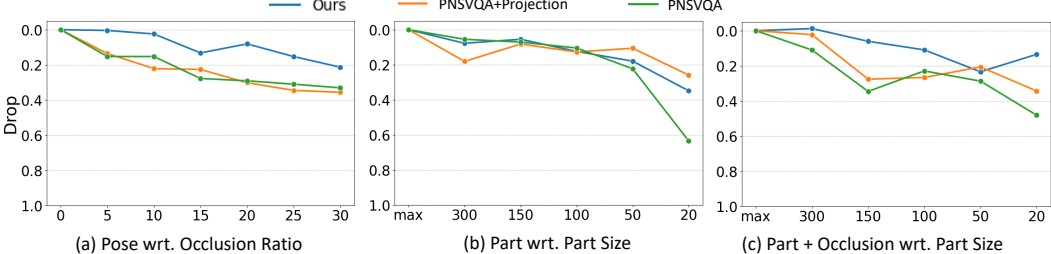

(a) Pose wrt. Occlusion Ratio        (b) Part wrt. Part Size        (c) Part + Occlusion wrt. Part Size

Figure 4: Analysis on questions of different difficulty levels. The plots show the relative accuracy drop of models, on pose questions w.r.t. different occlusion ratios (a), on part questions w.r.t. different part sizes (b), and on part+occlusion questions w.r.t. different part sizes (c).

**Questions with different occlusion ratios.** We sort pose-related questions into different sub-groups based on their *occlusion ratios* and evaluate the models on each of the sub-groups. The *occlusion ratio* $r$ of a question is the *minimum* of occlusion ratios for all the objects in its reasoning trace. We choose $r$ from $0\%$ to $30\%$, in increment of $5\%$. The results are shown is Fig. 4 (a). Our PO3D-VQA is much more robust to occlusions compared to the other two methods: while the performances of all

---

[3]Relative accuracy drop means the ratio of absolute accuracy drop and the original accuracy. For example, if a model's accuracy drops from 50% to 45%, its relative accuracy drop is 10%.

the three models decrease as the occlusion ratio increases, the relative drop of ours is much smaller than others. The results show that our render-and-compare scene parser is more robust to heavy occlusions compared with the discriminative methods.

**Questions with different part sizes.** Questions about small parts are harder than the ones about larger parts. We sort the questions into different part size intervals $(s, t)$, where the *largest* part that the question refers to has an area (number of pixels occupied) larger than $s$ and smaller than $t$. We compare the models on the part questions and the part+occlusion questions with different part sizes in Fig. 4 (b) and (c). In (b), the accuracy drop of PO3D-VQA is smaller than PNSVQA+Projection and PNSVQA when parts get smaller. In (c), PNSVQA+Projection is slightly better than our model and they are both better than the original PNSVQA.

In summary, by sorting questions into different difficulty levels based on occlusion ratios and part sizes, we show the advantage of our PO3D-VQA on harder questions, indicating that our model is robust to occlusions and small part sizes.

**Qualitative results.** Fig. 9 shows examples of predictions for our model and PNSVQA variants. In (a), the question asks about occlusion, but with a slight error in the pose prediction, PNSVQA+Projection misses the occluded bus and predicts the wrong answer, while our model is correct with **accurate pose**. In (b), the question refers to the heavily occluded minivan that is difficult to detect, but our model gets the correct prediction thanks to its **robustness to occlusions**.

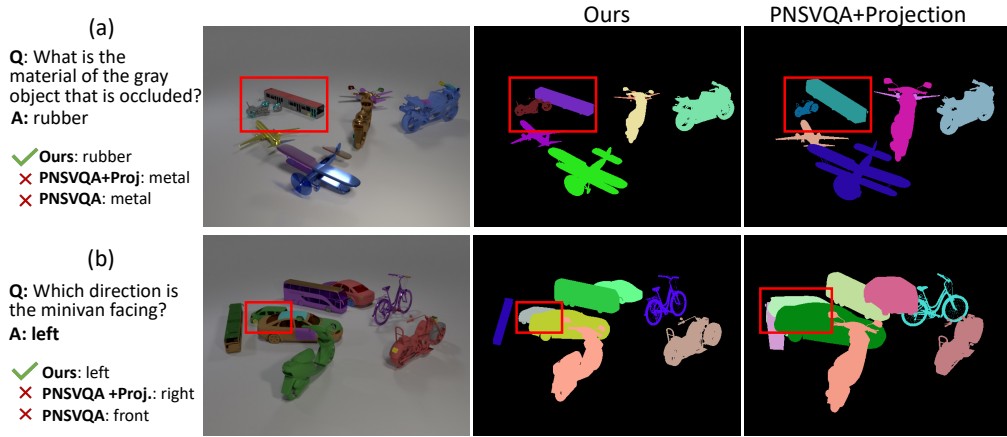

Figure 5: Examples of models' predictions. Our model (a) predicts the object pose accurately and (b) is robust to heavy occlusions. Red boxes are for visualization only.

**Limitations and failure cases.** Due to the difficulties of collecting real images with compositional scenes and 3D annotations, our work is currently limited by its synthetic nature. For PO3D-VQA, it sometimes fails to detect multiple objects if they are from the same category and heavily overlap (see Appendix D for more visualizations). 3D NMS can effectively improve the dense scene parsing results when objects are from different categories, but conceptually it is limited when objects are from the same category. However, 6D pose estimation in dense scenes is a challenging problem, whereas many current works on 6D pose estimation are still focusing on simple scenes with single objects [38, 50, 57].

## 6  Further Discussion

In this section, we discuss two meaningful extensions of our work: the incorporation of z-direction questions and the application of our model to real-world images.

**Z-direction questions**. While the proposed Super-CLEVR-3D dataset has been designed with 3D-aware questions, all objects within it are placed on the same surface. Introducing variability in the z direction can further enrich our dataset with more comprehensive 3D spatial relationships.

We consider the scenario where aeroplane category, is in different elevations, introducing the z dimension into the spatial relationships (see Fig. 6). This allowed us to formulate questions that probe the model's understanding of height relationships and depth perception. We create a subset containing 100 images and 379 questions and test our PO3D-VQA model directly on it without retraining the 6D

parser. On this dataset, our PO3D model achieves 90.33% accuracy on height relationship questions and 78.89% on depth-related questions, suggesting that our model can successfully handle questions about height. As the baseline models only use the bounding box to determine the spatial relationship between objects, they are not able to determine the height relationships.

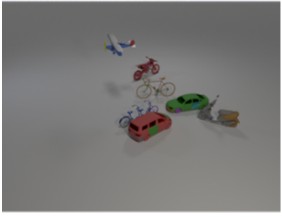 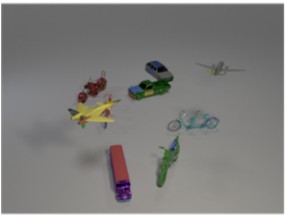

Figure 6: Example images and questions of objects with different elevations.

**Extension to real-world images** While our PO3D-VQA model has demonstrated impressive performance on the synthetic Super-CLEVR-3D dataset, an essential research direction is extending it to real images or other 3D VQA datasets (such as GQA and FE-3DGQA). However, it's not trivial to truly evaluate it on these real-world problems, and a primary challenge is the lack of 3D annotations and the highly articulated categories (like the human body) in these datasets.

However, we show that our PO3D-VQA model can, in principle, work on realistic images. We generate several realistic image samples manually using the vehicle objects (e.g. car, bus, bicycle) from ImageNet with 3D annotation (see Fig. 7) and real-image background. In this experiment, the pose estimator is trained on the PASCAL3D+ dataset, and is used to predict the poses of objects from the image before pasting, as shown in (b). The attribute (color) prediction module is trained on Super-CLEVR-3D and the object shapes are predicted by a ResNet trained on ImageNet. Our model can correctly predict answers to questions about the object pose, parts, and occlusions, e.g. "Which object is occluded by the mountain bike".

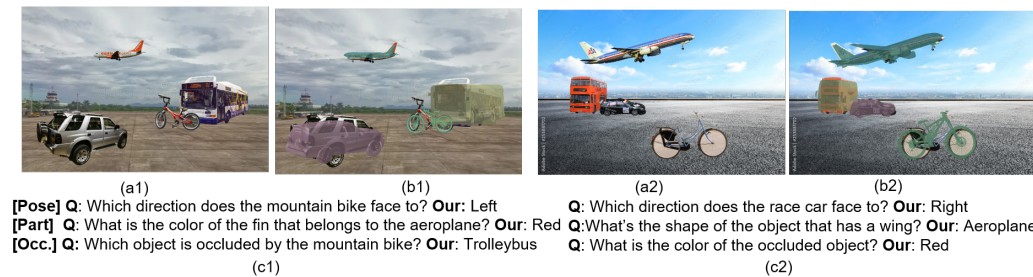

Figure 7: Examples of results on realistic images. Given a realistic image (a1, a2), our model can successfully estimate the 6D poses of objects (b1, b2) and answer the 3D-aware questions (c1, c2).

## 7 Conclusion

In this work, we study the task of 3D-aware VQA. We propose the Super-CLEVR-3D dataset containing questions explicitly querying 3D understanding including object parts, 3D poses, and occlusions. To address the task, a 3D-aware neural symbolic model PO3D-VQA is proposed, which enhances the probabilistic symbolic model with a robust 3D scene parser based on analysis-by-synthesis. With the merits of accurate 3D scene parsing and symbolic execution, our model outperforms existing methods by a large margin. Further analysis shows that the improvements are even larger on harder questions. With the dataset, the model, and the experiments, we highlight the benefit of symbolic execution and the importance of 3D understanding for 3D-aware VQA.

## Acknowledgements

We thank the anonymous reviewers for their valuable comments. We thank Qing Liu, Chenxi Liu, Elias Stengel-Eskin, Benjamin Van Durme for the helpful discussions on early version of the project. This work is supported by Office of Naval Research with grants N00014-23-1-2641, N00014-21-1-2812. A. Kortylewski acknowledges support via his Emmy Noether Research Group funded by the German Science Foundation (DFG) under Grant No.468670075.

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

# A Dataset Details

## A.1 Part list

In Super-CLEVR-3D, the parts of each objects are listed in Tab. 2

## A.2 Question templates

**Part Questions** we collect 9 part-based templates when generating the part-based questions, as shown in Tab. 4. In the table, <attribute> means one attribute from shape, material, color or size to be queried, <object> (or <object 1>, <object 2>) means one object to be filtered with a combination of shape, material, color, and size. Different from the pose and occlusion question, we don't query the size of the object.

**3D Pose questions** We design 17 3D pose-based templates in question generation (as shown in table 5). The 17 templates consist of: 1 template of the query of the pose; 4 questions of the query of shape, material, color, size, where the pose is in the filtering conditions; 12 templates about the query of shape, material, color, size, where the relationship of the pose is the filtering condition.

**Occlusion Questions** There are 35 templates in the occlusion question generation as shown in table 6, which consists of occlusion of objects and occlusion of parts.

The occlusion of objects consists of occlusion status and occlusion relationship. For the occlusion status of the object, there are 4 templates to query the shape, color, material, and size respectively. There are 2 occlusion relationships of objects (occluded and occluding), and each of them has 4 templates.

Similarly, we then create a template about occlusion status and occlusion relationship for the parts. The only difference between object and part is that the parts only have 3 attributes to be queried: shape (name), material and color.

## A.3 Statistics

As a result, a total of 314,988 part questions, 314,986 pose questions, and 228,397 occlusion questions and 314,988 occlusion questions with parts.

In Fig. 8, we show the distributions of all attributes of objects including categories, colors, sizes, and materials

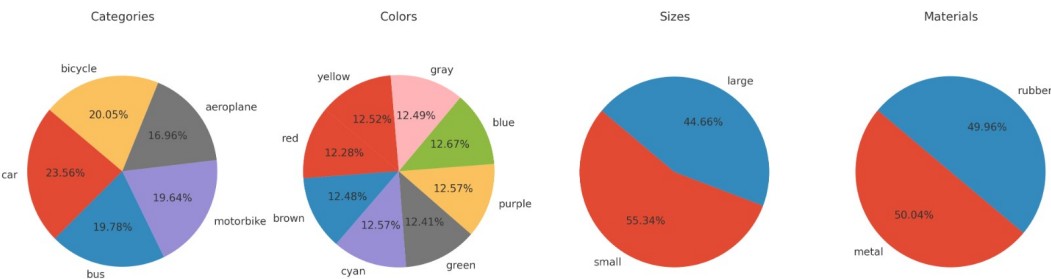

Figure 8: Distributions for all the attributes of objects including categories, colors, sizes, and materials

# B Implementation details for the baselines

The FiLM and mDETR are trained with default settings as in the official implementation. FiLM is trained for 100k iterations with batch size 256. mDETR is trained for 30 epochs with batch size 64 using 2 GPUs for both the grounding stage and the answer classification stage.

For P-NSVQA, we first train a MaskRCNN for 30k iterations with batch size 16 to detect the objects and parts, then train the attribute extraction model (using Res50 backbone) for 100 epochs with batch size 64. Different fully connected(FC) layers are used for a different type of question: the

part questions and occlusion questions have 4 FC layers for the shape, material, color, and size classification (as the parts also have size annotations in the dataset when generating scene files, but they are meaningless in the question answering). The pose question includes pose prediction of an object, so we add a new FC layer with 1 output dimension to predict the rotations, followed by an MSE loss during training. For different types of questions (part, pose and occlusion), the MaskRCNN and attribute extraction model are trained separately.

In the PNSVQA+Projection baseline, we first train a MaskRCNN to detect all of the objects and predict their 3D pose (azimuth, elevation and theta) without category labels in the scene. This MaskRCNN is trained with batch size 8 and iteration 15000. We use an SGD optimizer with a learning rate of 0.02, momentum of 0.9 and weight decay 0.0001. Then, we use the same setting as our PO3D-VQA to train a CNN to classify the attributes of objects and parts.

## C   Detailed results of Analysis

As an extension for section 5.4 in main paper, here we include the numerical value of accuracy and drop for the pose, part, occlusion + part question with reference to occlusion ratio or part size. The result is shown in Tab. 7, Tab. 9 and Tab. 8.

## D   Failure cases

Examples of failure cases of our PO3D-VQA, as described in Section 5.4 in main paper. In (a) and (b), PO3D-VQA misses the bicycle behind when two bicycles have a heavy overlap, the same for the two motorbikes in (c) and (d).

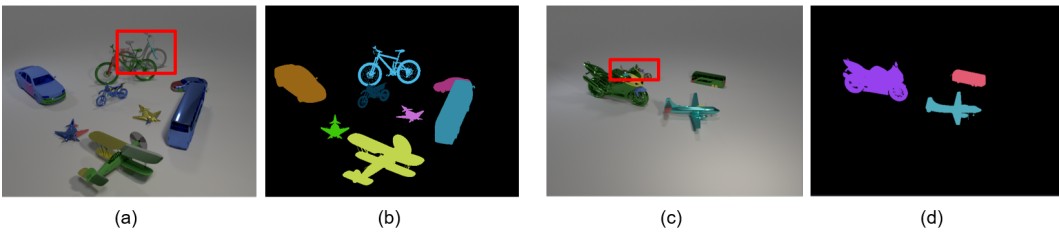

(a)                              (b)                              (c)                              (d)

Figure 9: Failure cases of our PO3D-VQA. (a) and (c) is the input images with the objects missed by the model. (b) and (c) is the re-projection results from the model.

Table 2: List of objects and parts.

| shape | part list |
| --- | --- |
| airliner | left door, front wheel, fin, right engine, propeller, back left wheel, left engine, back right wheel, left tailplane, right door, right tailplane, right wing, left wing |
| biplane | front wheel, fin, propeller, left tailplane, right tailplane, right wing, left wing |
| jet | left door, front wheel, fin, right engine, propeller, back left wheel, left engine, back right wheel, left tailplane, right tailplane, right wing, left wing |
| fighter | fin, right engine, left engine, left tailplane, right tailplane, right wing, left wing |
| utility bike | left handle, brake system, front wheel, left pedal, right handle, back wheel, saddle, carrier, fork, right crank arm, front fender, drive chain, back fender, left crank arm, side stand, right pedal |
| tandem bike | rearlight, front wheel, back wheel, fork, front fender, back fender |
| road bike | left handle, brake system, front wheel, left pedal, right handle, back wheel, saddle, fork, right crank arm, drive chain, left crank arm, right pedal |
| mountain bike | left handle, brake system, front wheel, left pedal, right handle, back wheel, saddle, fork, right crank arm, drive chain, left crank arm, right pedal |
| articulated bus | left tail light, front license plate, front right door, back bumper, right head light, front left wheel, left mirror, right tail light, back right door, back left wheel, back right wheel, back license plate, front right wheel, left head light, right mirror, trunk, mid right door, roof |
| double bus | left tail light, front license plate, front right door, front bumper, back bumper, right head light, front left wheel, left mirror, right tail light, back left wheel, back right wheel, back license plate, mid left door, front left door, front right wheel, left head light, right mirror, trunk, mid right door, roof |
| regular bus | left tail light, front license plate, front right door, front bumper, back bumper, right head light, front left wheel, left mirror, right tail light, back right door, back left wheel, back right wheel, back license plate, front right wheel, left head light, right mirror, trunk, mid right door, roof |
| school bus | left tail light, front license plate, front right door, front bumper, back bumper, right head light, front left wheel, left mirror, right tail light, back left wheel, back right wheel, back license plate, mid left door, front right wheel, left head light, right mirror, roof |
| truck | front left door, left tail light, left head light, back right wheel, right head light, front bumper, right mirror, front license plate, front right wheel, back bumper, left mirror, back left wheel, right tail light, hood, trunk, front left wheel, roof, front right door |
| suv | front left door, left tail light, left head light, back left door, back right wheel, right head light, front bumper, right mirror, front right wheel, back bumper, left mirror, back left wheel, right tail light, hood, trunk, front left wheel, back right door, roof, front right door |
| minivan | front left door, left tail light, left head light, back left door, back right wheel, right head light, front bumper, right mirror, front license plate, front right wheel, back bumper, left mirror, back left wheel, right tail light, hood, trunk, front left wheel, back right door, roof, front right door, back license plate |
| sedan | front left door, left tail light, left head light, back left door, back right wheel, right head light, front bumper, right mirror, front license plate, front right wheel, back bumper, left mirror, back left wheel, right tail light, hood, trunk, front left wheel, back right door, roof, front right door, back license plate |
| wagon | front left door, left tail light, left head light, back left door, back right wheel, right head light, front bumper, right mirror, front license plate, front right wheel, back bumper, left mirror, back left wheel, right tail light, hood, trunk, front left wheel, back right door, roof, front right door, back license plate |
| chopper | left handle, center headlight, front wheel, right handle, back wheel, center taillight, left mirror, gas tank, front fender, fork, drive chain, left footrest, right mirror, windscreen, engine, back fender, right exhaust, seat, panel, right footrest |
| scooter | left handle, center headlight, front wheel, right handle, back cover, back wheel, center taillight, left mirror, front cover, fork, drive chain, right mirror, engine, left exhaust, back fender, seat, panel |
| cruiser | left handle, center headlight, right headlight, right taillight, front wheel, right handle, back cover, back wheel, left taillight, left mirror, left headlight, gas tank, front cover, front fender, fork, drive chain, left footrest, license plate, right mirror, windscreen, left exhaust, back fender, right exhaust, seat, panel, right footrest |
| dirtbike | left handle, front wheel, right handle, back cover, back wheel, gas tank, front cover, front fender, fork, drive chain, left footrest, engine, right exhaust, seat, panel, right footrest |

Table 4: Templates of parts questions

| Templates | Count |
|---|---|
| What is the `<attribute>` of the `<part>` of the `<object>`? | 3 |
| What is the `<attribute>` of the `<object>` that has a `<part>`? | 3 |
| What is the `<attribute>` of the `<part 1>` that belongs to the same object as the `<part 2>`? | 3 |

Table 5: Templates of pose questions

| Templates | Count |
|---|---|
| Which direction the `<object>` is facing? | 1 |
| What is the `<attribute>` of the `<object>` which face to the `<O>`? | 4 |
| What is the `<attribute>` of the `<object 1>` that faces the **same direction** as a `<object 2>` | 4 |
| What is the `<attribute>` of the `<object 1>` that faces the **opposite direction** as a `<object 2>` | 4 |
| What is the `<attribute>` of the `<object 1>` that faces the **vertical direction** as a `<object 2>` | 4 |

Table 6: Templates of occlusion questions

| Templates | Count |
|---|---|
| What is the `<attribute>` of the `<object>` that is occluded? | 4 |
| What is the `<attribute>` of the `<object 1>` that is occluded by the `<object 2>` ? | 4 |
| What is the `<attribute>` of the `<object 1>` that occludes the `<object 2>`? | 4 |
| Is the `<part>` of the `<object>` occluded? | 1 |
| Which part of the `<object>` is occluded? | 1 |
| What is the `<attribute>` of the `<object>` whose `<part>` is occluded? | 4 |
| What is the `<attribute>` of the `<part>` which belongs to an occluded `<object>`? | 3 |
| What is the `<attribute>` of the `<part 1>` which belongs to the `<object>` whose `<part 2>` is occluded? | 3 |
| Is the `<part>` of the `<object 1>` occluded by the `<object 2>` | 1 |
| What is the `<attribute>` of the `<object 1>` whose `<part>` is occluded by the `<object 2>` ? | 4 |
| What is the `<attribute>` of the `<part>` which belongs to `<object 1>` which is occluded by the `<object 2>` | 3 |
| What is the `<attribute>` of the `<part 1>` which belongs to the same object whose `<part 2>` is occluded by the `<object 2>`? | 3 |

Table 7: Accuracy value and relative drop for pose questions wrt. occlusion ratio

|  | Occlusion Ratio | 0 | 5 | 10 | 15 | 20 | 25 | 30 |
|---|---|---|---|---|---|---|---|---|
| PNSVQA | Accuracy | 87.43 | 74.09 | 74.09 | 63.16 | 62.01 | 60.33 | 58.52 |
|  | Drop | 0.00% | 15.26% | 15.26% | 27.76% | 29.08% | 31.00% | 33.07% |
| PNSVQA + Projection | Accuracy | 86.30 | 74.61 | 67.20 | 66.78 | 60.26 | 56.52 | 55.56 |
|  | Drop | 0.00% | 13.54% | 22.13% | 22.62% | 30.17% | 34.51% | 35.63% |
| Ours | Accuracy | **86.43** | **86.05** | **84.32** | **75.00** | **79.44** | **73.22** | **67.98** |
|  | Drop | 0.00% | **0.44%** | **2.44%** | **13.22%** | **8.09%** | **15.28%** | **21.35%** |

Table 8: Accuracy value and relative drop for occlusion + part wrt. part size

|  | Part Size | max | 300 | 150 | 100 | 50 | 20 |
|---|---|---|---|---|---|---|---|
| PNSVQA | Accuracy | 58.18 | 54.98 | 54.05 | 52.09 | 45.20 | 21.28 |
|  | Drop | 0.00% | 5.49% | 7.10% | 10.47% | 22.31% | 63.43% |
| PNSVQA + Projection | Accuracy | 61.85 | 50.64 | 56.77 | 53.97 | 55.29 | 45.83 |
|  | Drop | 0.00% | 18.11% | 8.20% | 12.74% | **10.60%** | **25.89%** |
| Ours | Accuracy | **81.68** | **75.32** | **77.20** | **71.54** | **67.00** | **53.19** |
|  | Drop | 0.00% | **7.78%** | **5.49%** | **12.41%** | 17.97% | 34.88% |

Table 9: Accuracy value and relative drop for part wrt. part size

| | Part Size | max | 300 | 150 | 100 | 50 | 20 |
|---|---|---|---|---|---|---|---|
| PNSVQA | Accuracy | 57.31 | 51.00 | 37.50 | 44.18 | 40.85 | 29.73 |
| | Drop | 0.00% | 11.02% | 34.57% | 22.92% | 28.73% | 48.12% |
| PNSVQA + Projection | Accuracy | 58.89 | 57.54 | 42.64 | 43.20 | 46.73 | 38.67 |
| | Drop | 0.00% | 2.30% | 27.60% | 26.65% | **20.65%** | 34.34% |
| Ours | Accuracy | **64.04** | **64.80** | **60.16** | **57.03** | **49.05** | **55.41** |
| | Drop | 0.00% | **-1.19%** | **6.06%** | **10.94%** | 23.41% | **13.48%** |

