# OpenReview forum: "3D-Aware Visual Question Answering about Parts, Poses and Occlusions"
_NeurIPS.cc/2023/Conference — NeurIPS 2023 poster_

### Official Review · Reviewer_NU9U · 2023-07-06

**Soundness:** 3 good
**Presentation:** 2 fair
**Contribution:** 2 fair
**Rating:** 5
**Confidence:** 4

**Summary:**

This paper introduces the concept of 3D-aware Visual Question Answering (VQA) and presents a new dataset called "Super-CLEVR-3D". It also proposes a model called "PO3D-VQA" that combines probabilistic neural symbolic program execution with deep neural networks using 3D generative representations of objects. The experimental results show that the proposed model outperforms existing methods, but there is still a significant performance gap compared to 2D VQA benchmarks, highlighting the need for further research in 3D-aware VQA.

**Strengths:**

+ Originality: The authors propose a new dataset, Super-CLEVR-3D, that extends an existing 2D dataset with 3D-aware questions, introducing a novel benchmark for evaluating 3D-aware VQA models. The paper presents the PO3D-VQA model, which combines probabilistic neural symbolic program execution with deep neural networks using 3D generative representations of objects, providing a novel approach for addressing the 3D-aware VQA task.

+ Quality: Thorough experimental results demonstrate the superiority of the proposed model, PO3D-VQA, over existing methods.

+ Clarity: The paper provides clear motivation, defines the task of 3D-aware VQA, and describes the proposed dataset and model in a detailed and comprehensible manner.

+ Significance: The paper addresses the importance of understanding the 3D structure of visual scenes in VQA, introduces a new dataset, and showcases improvements in accuracy, advancing the field of VQA.


**Weaknesses:**

- The paper addresses the VQA problem in 3D scenes but only takes images as input. Why not use point cloud or multi-view images which are more suitable for 3D scenario?

- Where are the ground truth information of pose and occlusion come from? Are they included in the Super-CLVER dataset?

- The model design of PO3D-VQA is somewhat weak. It looks like the combination of Neural Meshes and P-NSVQA.

- I wonder about the performance of only using language and using oracle object representation ( ground truth class and pose label). In this way, we can show the reasoning ability of the model, or just accurate object detection is needed to solve this task.

- How the proposed model compared with the scene graph-based method, since the method first parses the image as scene representations. What is the advantage of the neural symbolic method against deep graph networks?

- There is no limited discussion on dataset limitations: While the Super-CLEVR-3D dataset is introduced as an extension of an existing dataset, the paper does not extensively discuss the limitations or potential biases of the dataset. Providing insights into the dataset construction process, potential challenges, and potential mitigations would strengthen the validity of the findings. And there is no limited scalability discussion: While the experimental results show improvements over existing methods, the paper does not thoroughly discuss the scalability of the proposed model. Understanding how the model's performance scales with larger and more complex scenes would be valuable for assessing its practical applicability.

**Questions:**

Have you considered analyzing the failure cases of existing 2D VQA models on 3D-aware questions? This could provide insights into the limitations of 2D reasoning and highlight the unique strengths of the 3D-aware approach.

Have you evaluated the performance of the model on larger-scale scenes or real-world datasets? If not, what challenges do you anticipate in scaling up the model to such scenarios?

**Limitations:**

Please refer to the weakness part.

---

> ### Author Rebuttal · Authors · 2023-08-08
>
> We appreciate the time and effort you put into reviewing our paper. Here we address the questions you raised.
>
> **Q1:  Why not use point cloud or multi-view images, instead of single-view images, as input?**
>
> A1: We would like to highlight the focus of our paper: 3D-aware VQA from 2D images, i.e. answering questions about 3D information that can be inferred from 2D images. We chose to use 2D images instead of 3D data because (1) In real-world applications, it is much harder to obtain point cloud or multi-view data compared with single-view 2D images. Therefore, understanding 3D information from 2D images is an important ability of computer vision models. (2) While point cloud or multi-view images may provide rich information, the three types of questions that we focus on in this paper, i.e. pose, parts, and occlusions, can be inferred from single-view images. Therefore, we argue that it is important and valuable to understand 3D information from 2D images. While 3D input data can be an extension of future work, this is beyond the scope of this paper. We would add this to the revision of the paper.
>
>
> **Q2: Where does the ground truth information of pose and occlusion come from? Are they included in the Super-CLEVR dataset?**
>
> A2: (a) The pose information is provided in scene annotation files of the Super-CLEVR dataset, which annotates the position, rotation (pose), and attributes of each object in the scene. (b) For the occlusion annotations, while they are not directly provided in the Super-CLEVR dataset, they can be easily computed by rerendering the scene based on the scene annotation files. To get the occlusion annotation, we render each object in the scene separately to get single-object images for each object which is guaranteed to be unoccluded, we compare the unoccluded masks in these rendered single-object images with the object masks of the multi-object images provided in Super-CLEVR. If the single-object mask differs from the multi-object by a threshold (15 pixels), we consider the object/part to be occluded. We will add the description to the revised paper.
>
> **Q3: The model design of PO3D-VQA is somewhat weak. It looks like the combination of Neural Meshes and P-NSVQA.**
>
> A3: We would like to politely disagree and argue that the model is not a simple concatenation of neural meshes and P-NSVQA. As discussed in Sec 4.2, non-trivial technical improvements have been made. Existing mesh-based pose estimation models [36, 47] are for single-class images: i.e., all objects in the image are from the same class and the object class is known during inference. However, in VQA settings, the scene contains multiple objects from multiple classes, posing a greater challenge to correctly classify the object and accurately solve the 6D pose. Therefore, we adopt the analysis-by-synthetic pipeline used in Neural Meshes and significantly extend the model by introducing including 3D-NMS, greedy proposal generation, and post-filtering are proposed in this paper to address the multi-class and  multi-object problem. In our paper, we also discuss the advantage of our method over the naive method that uses object detectors like FasterRCNN to detect and classify the object category.
>
> Moreover, besides the technical contributions, our work is the first to integrate 3D geometric neural meshes with VQA, which is shown to be effective and achieves strong results compared with existing methods. We would like to highlight this contribution and suggest the importance of 3D geometry in VQA.
>
> **Q4: Performance using oracle object representation, in order to show the reasoning ability of the model.**
>
> A4: Thank you for the suggestion. We replace the perception with the oracle object representation provided in the annotations and run the reasoning module (P-NSVQA) based on it. The model achieves 99% using the oracle representations, suggesting the strong ability of the symbolic reasoning module.
>
> **Q5: What is the advantage of the neural symbolic method against deep graph networks?**
>
> A5: We agree that graph models are widely adopted in VQA, which has been shown in previous works (Hu et al., 2019). However, compared with these models, neural symbolic methods like P-NSVQA or NSVQA have been shown to be more effective and achieve state-of-the-art performance on both the CLEVR dataset and the SuperCLEVR dataset. In addition, the modular symbolic methods have better interpretability (step-by-step reasoning), data efficiency, and robustness in the out-of-distribution settings, as shown in previous works [38, 31, 51].
>
> **Q6: Have you evaluated the performance of the model on larger-scale scenes or real-world datasets? If not, what challenges do you anticipate in scaling up the model to such scenarios?**
>
> A6: Please refer the the **general response** about results on the real images, as well as challenges for doing experiments on large-scale datasets.
>
> **Q7: Discussion of failure cases of baseline model.**
>
> A7: We show such examples in Figure 5 of the main paper, also here we provide two more examples in Fig-4 of the rebuttal PDF. The 2D models are hard to locate the small parts and determine if an object is occluded when the occlusion area is small. For the PNSVQA-Project method, it will make similar mistakes as the pose estimator is not precise enough.
>
> **Q8: Discussion of the dataset limitations.**
>
> A8: Regarding potential biases in the dataset - As the original Super-CLEVR dataset is designed to study VQA domain generalization, the distribution of the objects/attributes, question redundancy, and the compositionality of concepts are balanced. We built our Super-CLEVR-3D on the $default$ version of Super-CLEVR, where the concepts are well-balanced. We will include more statistics about the dataset distribution in the supplementary materials.
>
> [1] Hu, Ronghang, et al. "Language-conditioned graph networks for relational reasoning." Proceedings of the IEEE/CVF international conference on computer vision. 2019.

---

### Official Review · Reviewer_SUrx · 2023-07-07

**Soundness:** 4 excellent
**Presentation:** 4 excellent
**Contribution:** 4 excellent
**Rating:** 9
**Confidence:** 2

**Summary:**

1, The paper extends 2D VQA to the task of 3D-aware VQA, which requires the understanding of the 3D structure of visual scenes and includes knowledge of 3D object pose, parts, and occlusions.
2. Introducing the Super-CLEVR-3D dataset, which contains questions about object parts, their 3D poses, and occlusions.
3. The proposed model, PO3D-VQA, is a 3D-aware VQA model that has two key techniques: probabilistic neural symbolic program execution, and deep neural networks paired with 3D generative object representations for robust visual recognition.

**Strengths:**

1. Proposed an interesting 3D-aware VQA task.
2. Introduced a valuable Super-CLEVR-3D dataset.
3. Developed a novel  PO3D-VQA method, that significantly outperforms prior methods.
4. Detailed mathematical explanation of the components of the methods.
5. Comprehensive analysis and discussions.

**Weaknesses:**

1. Lack of some statistics about the constructed Super-CLEVR-3D dataset.

**Questions:**

1. What type of parts are included in the part questions?
2. Is there a threshold for an object to be considered occluded?

**Limitations:**

1. As described in the paper, the work is currently limited by synthetic scenes.
2. As described in the paper, the method is sensitive to pose prediction errors.

---

> ### Author Rebuttal · Authors · 2023-08-08
>
> We appreciate your thoughtful and positive feedback on our paper. Please find our responses below:
>
> **Q0 & A0: Please refer to the general response for generalization to real images.**
>
> **Q1: Lack of some statistics about the constructed Super-CLEVR-3D dataset.**
>
> A1: We include the dataset statistics, as well as the list of question templates and the object-part list in the supplementary materials. We will revise and include important statistics in the main paper, especially the distribution of the questions, objects and attributes (classes, shape, material, color, size), parts, and occlusion ratios of objects and parts. Here we list the distribution of all attributes of objects in these tables.
>
> |           | car     | bus     | motorbike | aeroplane | bicycle  |
> |-----------|---------|---------|-----------|-----------|----------|
> | Percentage | 23.56%  | 19.78%  | 19.64%    | 16.96%    | 20.05%   |
>
> |           | red     | brown   | cyan    | green   | purple  | blue    | gray    | yellow  |
> |-----------|---------|---------|---------|---------|---------|---------|---------|---------|
> | Percentage| 12.28%  | 12.48%  | 12.57%  | 12.41%  | 12.57%  | 12.67%  | 12.49%  | 12.52%  |
>
> |           | small   | large   |
> |-----------|---------|---------|
> | Percentage| 55.34%  | 44.66%  |
>
>
> |           | metal   | rubber  |
> |-----------|---------|---------|
> | Percentage| 50.04%  | 49.96%  |
>
> **Q2: What type of parts are included in the part questions?**
>
> A2: Thanks for bringing this up and we will include more details in our revised paper. The object parts are mined from the UDA-Part dataset [32], where each 3D object model is annotated with parts. We cleaned the UDA-Part annotations by removing the noisy annotations and the extremely small parts which can hardly be seen when rendered onto images. The final list of parts for each object type is included in the supplementary materials.
>
> **Q3: Is there a threshold for an object to be considered occluded?**
>
> A3: Yes, there is a threshold. When generating the questions, we consider an object to be “not occluded” if the number of the occluded pixels is zero; we consider an object to be ”occluded” when the number of occluded pixels is larger than 15 (with 640 by 480 image resolution). We will include the threshold in the implementation details.

---

### Official Review · Reviewer_HP3a · 2023-07-08

**Soundness:** 3 good
**Presentation:** 3 good
**Contribution:** 3 good
**Rating:** 6
**Confidence:** 2

**Summary:**

This work introduces a framework to Visual Question Answering (VQA) that incorporates understanding of the 3D structure of scenes, a significant leap from traditional 2D-based models. To evaluate the task, they proposed a new dataset, Super-CLEVR-3D, designed specifically for 3D-aware VQA, containing questions that necessitate compositional reasoning regarding the 3D object parts, poses, and occlusions. To tackle these queries, they also put forth a new model, PO3D-VQA, combining probabilistic neural symbolic program execution for reasoning and deep neural networks for robust 3D scene parsing. Experiments show the proposed PO3D-VQA model outperforms existing techniques, especially on more complex questions, underscoring the need for 3D understanding in future VQA research. Despite the improvements, the noticeable performance gap compared to 2D VQA benchmarks indicates that 3D-aware VQA remains a critical area of exploration.






**Strengths:**

I believe the studied direction, 3D-aware 2D-VQA is important for our community, especially when we want to tackle real-world tasks. Both the proposed VQA dataset and the framework, PO3D-VQA, takes a good attempt towards the big goal.

All the used techniques as shown in method and Figure 2 are sound and composed in a reasonable way. Overall, the paper is easy to follow.

Results in Table 1 demonstrated the strength of the proposed method, which yield significant improvements, and also indicate there are more to do in the future. Some ablation studies are included in Section 5.4.

**Weaknesses:**

As the paper claimed a 3D-VQA dataset contribution, I am curious how the model transfer performance from the dataset to real-images. For example, taking a picture with multiple real 3d models  and run the model to check the sim-to-real performance.

Also, an odd part is that the current motivation for this paper is 3D navigation and manipulation, but few of the defined problems are related to navigation or manipulation.
- Can the proposed model accurately locate parts of 3D objects that can help manipulate?
- Are there questions directly related to navigation and manipulation in 3D space?

The current question-answers in Figure1 in my mind is more related to previous VQA while less related to 3D VQA that really matters.

Besides, there are no failure cases understanding and analysis. For the proposed pipeline, the failed cases should include both the failure of perception module and the failure of reasoning module. Providing failure cases can help people understand the limitation of the proposed system, while not weaken the contribution.

**Questions:**

Please address the concerns raised above.

**Limitations:**

It seems the current framework cannot correct itself if the perception model generated wrong outputs.

---

> ### Author Rebuttal · Authors · 2023-08-08
>
> Thank you for your feedback. Here we address the questions:
>
> **Q0 & A0: Please refer to the general response for generalization to real images.**
>
> **Q1: About 3D navigation and manipulation: Are there questions directly related to navigation and manipulation in 3D space?**
>
> A1: There are no questions about navigation and manipulation in our dataset and we will clarify this in our final paper. In the submission, we talked about 3D navigation and manipulation because this is one of the motivations as well as the long-term goals of our work. By introducing the 3D VQA we make an initial step in that direction by designing 3D questions that the current 2D baseline models are not able to answer well. We also note that for manipulation and navigation, we would need to annotate parts that are useful for these interactions, which is a direction that we plan to pursue in future work.
>
> **Q2: Can the model accurately locate parts of 3D objects that can help manipulate?**
>
> A2: In our 3D VQA setting, the model can locate the part accurately in most cases. In Fig.3 of the rebuttal PDF, we visualize the part localization results predicted by our model in the SuperCLEVR-3D dataset. But as we stated in Q1, our 3D VQA dataset is just a first step, and the parts needed for 3D manipulation or navigation might be different from those annotated in our dataset.
>
> **Q3: The current question-answers in Figure1 in my mind is more related to previous VQA while less related to 3D VQA that really matters.**
>
> A3: As stated in our paper, the question in the dataset requires a comprehensive understanding of the 3D structure of scenes, like the 3D poses, occlusion relationship, and the hierarchical relationship between objects and parts. From our experiment, we show that the 2D baseline methods cannot handle this question without the 3D understanding. In this rebuttal, we add the z-direction relationship and depth-related questions on new images, where the 2D models are not able to determine the precise height (in the z-direction) and depth (see Fig-2 in rebuttal PDF).
>
> **Q4:  Failure case analysis: does the errors come from the perception module or the reasoning module?**
>
> A4: A4: We check the failure cases and provide two visualizations in Fig-5 of the rebuttal PDF. We observe that most of the failure cases are due to the errors of the 6D pose estimator, typically when multiple objects from the **same class** are overlapping. As the mesh-based 6D pose estimator detects objects and their 6D poses by class (Fig-3 II of the main paper), it may fail to parse all objects from the same category when they are too close to each other. 3D NMS can effectively improve the dense scene parsing when objects are from different categories, but conceptually it cannot help when objects are from the same categories in a dense scene. We argue that 6D pose estimation in dense scenes is still a challenging problem and many current works on 6D pose estimation still focus on simple scenes with single objects (Ma et al. 2022; Yu et al. 2014; Yang et at. 2022).
>
> **Q5: It seems the current framework cannot correct itself if the perception model generates wrong outputs.**
>
> A5: As our reasoning module is a probabilistic neuro-symbolic method (P-NSVQA [31]), it allows the model to output correct answers even when the vision module makes errors. We also introduce new probabilistic reason functions for filtering poses and occlusion in the reasoning module. As studied in the [31], with the same perception module, P-NSVQA can outperform the NSVQA in all test sets, which means it revises the errors from the vision predictions. We believe that the probabilistic reasoning module can also help our 3D perception model to revise some errors. But as with most modular VQA models, the errors from the vision model can inevitably impact the final model's accuracy and performance, which is also good for interpretability.
>
> [1] Ma, Wufei, et al. "Robust category-level 6d pose estimation with coarse-to-fine rendering of neural features." European Conference on Computer Vision. Cham: Springer Nature Switzerland, 2022.
>
> [2] Xiang, Yu, Roozbeh Mottaghi, and Silvio Savarese. "Beyond pascal: A benchmark for 3d object detection in the wild." IEEE winter conference on applications of computer vision. IEEE, 2014.
>
> [3] Ze, Yanjie, and Xiaolong Wang. "Category-level 6d object pose estimation in the wild: A semi-supervised learning approach and a new dataset." Advances in Neural Information Processing Systems 35 (2022): 27469-27483.

---

> > ### Comment · Reviewer_HP3a · 2023-08-14
> > **Reviewer response**
> >
> > I thank the efforts for making the rebuttal. I encourage the author to include all the discussion here in the revision especially for the real-image and failure cases. It would be better to include some real cases to show your pipeline can correct answer when vision module outputs errors.

---

> > > ### Author Response · Authors · 2023-08-21
> > > **Thanks for your comments**
> > >
> > > Thanks for your comments. We will add the discussion about real-image and failure cases into our revised paper. Also, we will add some cases to show how the error from the vision module can be corrected by the probabilistic reasoning module.

---

### Official Review · Reviewer_sJrg · 2023-07-09

**Soundness:** 2 fair
**Presentation:** 3 good
**Contribution:** 3 good
**Rating:** 6
**Confidence:** 5

**Summary:**

This paper is seeks to improve the VQA models' understanding of 3D structure of images, particularly parts, poses, and occlusions.  There are two main contributions:
1. Super-CLEVR-3D: a dataset that contains questions about parts, poses, and occlusions.
2. PO3D-VQA: a 3D aware VQA model that combines nuerosymbolic program execution for reasoning and 3D generative representations.

Experimental settings: the proposed model is tested on the proposed dataset Super-CLEVR-3D.

**Strengths:**

1. The dataset is a good contribution for studying 3D vision-language reasoning.
2. The proposed model, PO3D-VQA is well-explained and motivated, along with the addition of a 6D parsing module and 3D NMS for better scene understanding.
3. The proposed model is a good proof-of-concept for the combination of neurosymbolic program execution and features learned using deep learning.
4. Relevant baselines are used and the experimental setting is soundand well explained.

**Weaknesses:**

1. The proposed model is only tested on the Super-CLEVR-3D dataset
2. There aren't any questions about "z" direction i.e. above/below relationships as the objects in [31] are always on the same surface.  Because of this it is questionable to call the dataset "3D" as only 2D relationships are covered. The dataset is missing templates/questions about depth, distance between objects etc.
4. Super-CLEVR-3D dataset only 5 categories, all of which are forms of ‘vehicles’. This could also lead to an overestimation of the object detection module that is used in the model.
5. The method could be tested on several other 3D aware datasets: for instance GQA [Hudson et al CVPR 2019], or this work https://arxiv.org/abs/2209.12028. Also see Q2, Q3, Q4 for more questions about evaluations.

**Questions:**

1. Has jointly training the pose estimator and attribute classifier been explored? Could training them jointly limit the erroneous pose prediction instances?
2. How is the performance on the original Super-CLEVR dataset or on the CLEVR dataset?
3. Does the method generalize well to cases where parts/poses/occlusions are not mentioned? Can the proposed model be reliably used for VQA outside of the Super-CLEVR setting?
4. Can this model be used for real-world images with questions about parts/poses/occlusions?

**Limitations:**

Limitations are discussed [line 348]. Discussions along the lines of Q2/3/4 above would also be useful when describing the limitations.

---

> ### Author Rebuttal · Authors · 2023-08-08
>
> We sincerely thank you for the detailed discussions and the appreciation of our work. We would like to address the concerns and questions you raised:
>
> **Q0 & A0: Please refer to the general response for generalization to real images.**
>
> **Q1: There aren't questions about “z” direction (“above/below” relationships) as the objects are always on the ground.**
>
> A1: We thank the reviewer for the suggestion. As suggested, we create new images where some airplanes are flying in the air (while the cars are kept on the ground plane). Examples of the images are shown in Fig-2 of the rebuttal PDF. For the new images, we ask “z-questions” about height relationships and depth. To be more specific, the height relationship asks about “above/below” relationships that can be used to query the objects (e.g. “What shape is the cyan thing above the blue shiny car?”); for the depth question we add the comparison of depth between two objects where the size and bounding box location is not sufficient for the prediction (e.g. “Is the aeroplane closer to the camera than the school bus?”). In this way, we create a subset containing 100 images and 379 questions and test our PO3D model on this subset.
>
> We test the PO3D-VQA model directly on the new dataset without retraining. On this dataset, our PO3D model achieves 90.33% accuracy on height relationships questions and 78.89% accuracy on the distance-based question suggesting that our model is not limited to objects on the ground plane and can successfully handle questions about height. As the baseline models only use the bounding box to determine the spatial relationship between objects, they are not able to determine the height relationships.
>
>
> **Q2: Super-CLEVR-3D dataset only 5 categories, all of which are forms of vehicles.**
>
> A2: Our model is trained from 3D CAD models that have detailed part annotations. Such well-annotated CAD models are difficult to obtain, therefore we are limited to the object categories covered in the UDA-Part dataset [32]. While there are large-scale 3D object models like Objaverse [A] defining and annotating parts on these meshes is challenging and time-consuming. However, we argue that the 5 categories are already very challenging for existing models, as suggested in [31, 32]. In our work, we not only improve the performance greatly over the existing methods but also extend the dataset with different types of questions, which is a non-trivial contribution.
>
> **Q3: Could jointly training the pose estimator and attribute classifier bring better results?**
>
> A3: Thanks for the suggestion. We agree that it’s a valuable direction to study if the pose estimator can benefit from the joint training. However, there remains a non-trivial technical gap that the pose estimation process is an iterative render-and-compare process that cannot be easily integrated with an attribute classification head. Based on the significant amount of effort required, we leave this as future work.
>
>
> **Q4: How is the performance on the original Super-CLEVR dataset or on the CLEVR dataset?**
>
> A4: We test our model on the original Super-CLEVR dataset. The accuracy of the PO3D-VQA model is 88.40%, which is lower than the SOTA methods (95% PNSVQA). Given that the original dataset primarily focuses on 2D questions, precision in 2D location becomes crucial. While MaskRCNN can train the detection directly from the bounding boxes supervision, our 6D pose estimator is trained to solve 6D pose but not explicitly designed for 2D detection. In Fig-5 of the rebuttal pdf, we show that our model will miss some objects especially when multiple objects are too close or overlapping.  As the mesh-based 6D pose estimator detects the object's class by class (Fig-3 II of the main paper), it will fail to parse all the objects from one feature map of one category in such cases.  We will add this limitation analysis to the paper.
>
> **Q5: Does the method generalize well to cases where parts/poses/occlusions are not mentioned or VQA outside of the Super-CLEVR setting?**
>
> A5: First, the original Super-CLEVR dataset does not contain parts/poses/occlusions questions. Our model achieves 88.40% accuracy on the original Super-CLEVR, suggesting that the model generalizes well to this setting. Second, as mentioned in the general response, the model works reasonably well on more realistic images, which indicates that our model is not limited to the Super-CLEVR setting.
>
> [A] Objaverse: A Universe of Annotated 3D Objects. arXiv 2212.08051

---

> > ### Comment · Reviewer_sJrg · 2023-08-15
> > **thank you for the rebuttal**
> >
> > Thank you for your detailed rebuttal and answers to the reviewers' questions.
> >
> > - The real-world experiments are interesting and should be added to the main paper even though they are preliminary / not full-scale.  And I appreciate the authors' response about why scaling it to a large realistic dataset would be challenging.
> > - Thanks for the "z-direction" experiments -- do you plan to add the new subset that you produced during the rebuttal to the proposed dataset?
> > - The fact that performance drops for the older Super-CLEVR dataset is problematic -- perhaps the models may be overfitting to either dataset?

---

> > > ### Author Response · Authors · 2023-08-21
> > > **Thanks for your comments**
> > >
> > > 1. Thanks. We will add a new section with a detailed description and analysis of the real-world experiments following our discussion in the rebuttal.
> > >
> > >
> > > 2. Yes, we are working on expanding the "z-direction” dataset with more images as an independent subset in our dataset. And we will add these experiments into our main paper.
> > >
> > >
> > > 3. Thanks for your comment. On the old SuperCLEVR dataset, the inferior performance of PO3D-VQA is mainly due to the quality of the predicted 2D bounding boxes, as our model is not specifically trained with bounding box supervision, while the PNSVQA model directly builds on a  MaskRCNN which is specifically designed for 2D detection and hence gives a significant performance boost in 2D questions. In contrast, we extended our PO3D-VQA model to  2D questions, by predicting the 2D boxes after first inferring the 3D scene parameters,  which is arguably much more challenging compared to plain 2D prediction of bounding boxes. Particularly for dense scenes with multiple overlapping objects (as shown in the failure cases in our rebuttal), our PO3D-VQA model sometimes misses highly occluded objects, and these missing objects lead to the performance drop of PO3D-VQA.
> > >
> > >     In order to demonstrate that the missing objects in dense scenes account for most of the performance gap, we conduct a new experiment. We compare the predicted object boxes with the groundtruth boxes using 2D IoU to detect  ‘missing’ object. Specifically, missing objects in the ground truth annotation do not overlap with any predicted object box  (i.e. with IoU over 0.0). Note that this is a very restrictive setting since we only flag those objects as missing that do not have any overlap with any of the detected boxes. Nevertheless, we find that by adding back these missing objects, our PO3D-VQA achieves a performance of 94.3%, which is comparable with the previous SOTA. This experiment shows that the missing objects as demonstrated in the failure cases is also the main reason of the performance gap between PO3D-VQA and PNSVQA on the old SuperCLEVR.
> > >
> > >     We would like to point out that our PO3D-VQA parses the scene from a 3D perspective and is specifically designed for 3D-based questions. Evaluating the model on SuperCLEVR is not necessarily a fair comparison as it doesn’t use the 2D bounding box supervision in the older SuperCLEVR. In future work we aim to improve the PO3D-VQA in dense scenes or a hybrid neural-symbolic model that excels in both 2D and 3D questions.

---

### Author Rebuttal · Authors · 2023-08-08

# General Response

We thank all the reviewers for the feedback. We are glad that all four reviews are positive and acknowledge both of our contributions: (1) The Super-CLEVR-3D dataset, i.e. a new VQA dataset that introduces an important new task (HP3a, NU9U) that requires reasoning over object parts, 3D poses, and occlusions, and (2) the originality of our proposed PO3D-VQA model (NU9U, SUrx, sJrg) for solving these tasks. We are happy that the reviewers highlighted the significance of our experimental improvements over the baselines (HP3a, SUrx, NU9U), and the clear motivation and good writing quality of the paper.

Here we provide a general response about how our proposed PO3D-VQA model can be extended beyond Super-CLEVR-3D to datasets that contain real-world imagery.

First, **we agree that our work should be extended to real images or other 3D VQA datasets as an important research topic, and we have started doing this in the current ongoing work**. However, we note that extending such a 3D-aware VQA to datasets is non-trivial. For the real-world data, existing real-world VQA datasets (like GQA and FE-3DGQA suggested by reviewer-sJrg) lack 3D annotations to objects and 3D-related question-answers (questions in existing VQA datasets refer mostly to the 2D space), which makes it hard to train and test our model on them. To enable 3D-aware VQA on these datasets, we need to generate additional 3D annotations on them. Moreover, real VQA datasets contain a large variety of object classes with many being highly articulated (e.g. humans and animals). We are working on annotating these datasets with 3D and scaling up the number of object classes in our PO3D-VQA model, which could allow us to train and test our model, but this requires a significant amount of effort and time.

However, to show that our PO3D-VQA model can, in principle, work on realistic images, we tested it on several more realistic image samples that were generated with objects from real datasets that had 3D annotations given (see Fig-1 of rebuttal file). This small preliminary experiment shows that our model can successfully answer 3D-related questions on realistic images.

To give more details on this experiment: The example images are manually created using the vehicle objects (e.g. car, bus, bicycle) from ImageNet. In this experiment, the pose estimator is trained on the PASCAL3D+ dataset, which can successfully predict the poses of objects in the image, as shown in (b). The attribute (color) prediction module is trained on Super-CLEVR-3D and the object shapes are predicted by a ResNet trained on ImageNet. Our model can correctly predict answers to questions about the object pose, parts, and occlusions, e.g. “Which object is occluded by the mountain bike”.

From the provided examples, we hope to demonstrate that the difficulty of annotation makes it hard to use vision algorithms that are 3D-aware, not just our model. Additionally, we also conduct experiments on new types of questions with more complex scenes as asked by some reviewers. We hope these can help to address the reviewers’ questions about our synthetic dataset and our proposed 3D-aware VQA model.

---

### Decision · Program_Chairs · 2023-09-21

**Decision:**

Accept (poster)

**Comment:**

Four experts reviewed this paper with all accepted recommendations. The area chairs agree that this work makes a very important contribution by introducing a new 3D-aware VQA dataset. The reviewers did raise some valuable concerns that should be addressed in the final camera-ready version of the paper. It would be also nice if the authors could include some discussions on how to make progress on this challenge by leveraging the recent success of 3D foundational models (e.g. [1, 2, 3] ) in the final version.

[1] 3D Concept Learning and Reasoning from Multi-View Images. CVPR 2023

[2] Scalable 3D Captioning with Pretrained Models.

[3] 3D-LLM: Injecting the 3D World into Large Language Models